# Use of minimally invasive tissue sampling to determine the contribution of diarrheal diseases to under-five mortality and associated co-morbidities and co-infections in children with fatal diarrheal diseases in Africa and Bangladesh

Portia Chipo Mutevedzi[1]\*, Zachary J. Madewell[2]\*, Karen L. Kotloff[3], Quique Bassat[4,5,6,7,8], Percina Joao Chirinda[5], Anelsio C. A. Cossa[5], Elisio G. Xerinda[5], Victor Akelo[9], Paul K. Mitei[10], Elizabeth Oele[11], Richard Omore[9], Dickens Onyango[11], Joseph Bangura[12], Ronita Luke[12], Andrew Moseray[13], Ikechukwu Udo Ogbuanu[14], Tom Sesay[13], Nega Assefa[15,16], Temesgen Teferi Libe[15], Lola Madrid[15,16], Melisachew M. Yeshi[17], J. Anthony G. Scott[16], Nelesh P. Govender[18], Sanjay G. Lala[19], Shabir A. Madhi[20,21], Sana Mahtab[20], Adama Mamby Keita[22], Doh Sanogo[22], Samba O. Sow[22], Milagritos D. Tapia[3], Shams El Arifeen[23], Emily S. Gurley[23,24], Beth A. Tippett Barr[25], Cynthia G. Whitney[1], Dianna M. Blau[2], Inacio Mandomando[4,5,26], for the CHAMPS Consortium¶

1 Emory Global Health Institute, Emory University, Atlanta, Georgia, United States of America, 2 Global Health Center, Centers for Disease Control and Prevention, Atlanta, Georgia, United States of America, 3 Department of Pediatrics and Department of Medicine, Center for Vaccine Development and Global Health, University of Maryland School of Medicine, Baltimore, Maryland, United States of America, 4 ISGlobal–Hospital Clínic, Universitat de Barcelona, Barcelona, Spain, 5 Centro de Investigação em Saúde de Manhiça–CISM, Maputo, Mozambique, 6 ICREA, Barcelona, Spain, 7 Pediatrics Department, Hospital Sant Joan de Déu, Universitat de Barcelona, Barcelona, Spain, 8 CIBER de Epidemiología y Salud Pública, Instituto de Salud Carlos III, Madrid, Spain, 9 Kenya Medical Research Institute, Center for Global Health Research, Kisumu, Kenya, 10 Kisumu East District Hospital, Kisumu, Kenya, 11 Kisumu County Department of Health, Kisumu, Kenya, 12 Ministry of Health and Sanitation, Freetown, Sierra Leone, 13 Department of Public Health, School of Community Health Sciences, Njala University Bo Campus, Bo City, Sierra Leone, 14 Crown Agents, Freetown, Sierra Leone, 15 College of Health and Medical Sciences, Haramaya University, Harar, Ethiopia, 16 Department of Infectious Disease Epidemiology, London School of Hygiene & Tropical Medicine, London, United Kingdom, 17 Department of Pathology, Ayder Referral Hospital, Mekelle, Ethiopia, 18 Mycology Reference Unit, National Institute for Communicable Diseases, Johannesburg, South Africa, 19 Department of Paediatrics & Child Health, Faculty of Health Sciences, University of the Witwatersrand, Johannesburg, South Africa, 20 South African Medical Research Council Vaccines and Infectious Diseases Analytics Research Unit, University of the Witwatersrand, Johannesburg, South Africa, 21 Wits Infectious Diseases and Oncology Research Institute, Faculty of Health Sciences, University of the Witwatersrand, Johannesburg, South Africa, 22 Centre pour le Développement des Vaccins, Ministère de la Santé, Bamako, Mali, 23 International Center for Diarrhoeal Diseases Research, Dhaka, Bangladesh, 24 Department of Epidemiology, Johns Hopkins Bloomberg School of Public Health, Baltimore, Maryland, United States of America, 25 Nyanja Health Research Institute, Salima, Malawi, 26 Instituto Nacional de Saúde, Ministério de Saúde, Maputo, Mozambique

¶ Membership of the CHAMPS Consortium is listed in the Acknowledgments.
\* ock0@cdc.gov (ZJM); pmuteve@emory.edu (PCM)

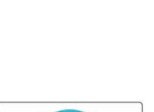

## Abstract

Achieving the Sustainable Development Goal of reducing child mortality to <25 deaths per 1000 live births by 2030 requires strategies to prevent diarrheal disease-related morbidity and mortality. Accurate etiological diagnosis is essential. This study

**Data availability statement:** CHAMPS data are open access and datasets are available for download or request on www.champshealth. org.

**Funding:** CHAMPS is funded by The Bill & Melinda Gates Foundation (OPP1126780 to CGW) which provided input into site selection decisions and methodology and scope of CHAMPS. The funders had no role in study design, data collection and analysis, decision to publish, or preparation of the manuscript. None of the authors receive a salary from any of the funders.

**Competing interests:** The authors have declared that no competing interests exist.

used postmortem diagnostics to investigate the contribution of diarrhea to under-5 mortality and examine co-morbidities and co-infections in Africa and South Asia. Child Health and Mortality Prevention Surveillance (CHAMPS) generates data on child deaths through minimally invasive tissue sampling, clinical record review, and verbal autopsies. Multidisciplinary panels assign cause(s) of death using WHO International Classification of Diseases. This analysis included deaths among children aged 1–59 months enrolled from 18 December 2016–31 December 2023 across six African sites (Ethiopia, Mali, Kenya, Sierra Leone, Mozambique, South Africa) and Bangladesh. Of 1517 deaths assessed, diarrhea was in the causal pathway in 240 (15.8%). The proportion of diarrhea-related deaths was highest in Ethiopia (41.0%, 34/83), followed by Bangladesh, (30.0%, 3/10), Mozambique (21.7%, 56/258), Mali (17.5%, 18/103), Kenya (13.9%, 51/366), Sierra Leone (12.8%, 46/358), and South Africa (9.4%, 32/339). Diarrhea was underlying cause in 44.2% (106/240) of cases and immediate/antecedent cause in 58.3% (140/240), with some deaths involving multiple roles in the causal chain. When diarrhea was underlying cause, sepsis (33.0%) and lower respiratory infections (25.5%) were common downstream conditions; when an antecedent/immediate cause, leading underlying causes were malnutrition (64.3%) and HIV (13.6%). No pathogen was identified in 49.6% (119/240) of diarrhea-related deaths; among these, diarrhea was underlying cause in 42.9%. Among the 121 pathogen-attributed deaths, the most frequent were EAEC (34.7%), typical EPEC (15.7%), Shigella/EIEC (14.0%), ST-ETEC (12.4%), rotavirus (26.4%), and adenovirus (non-40/41: 19.0%; 40/41: 5.0%). These pathogens were frequently identified as co-infections. Diarrheal disease accounted for a substantial share of child deaths across CHAMPS sites. Reducing mortality will require preventing diarrhea and addressing key contributors such as malnutrition and HIV.

## Introduction

In 2023, the United Nations Inter-agency Group identified diarrheal diseases as the third leading cause of under-5 mortality globally (9%), followed by malaria as the fourth (9%), exceeded only by prematurity (18%), lower respiratory tract infections (14%), and birth asphyxia/trauma (12%) [1]. There has been progress in reducing childhood diarrhea-attributed deaths worldwide, with a 60.3% decline observed from 1990 to 2021, decreasing from 2.93 million (95% UI: 2.31–3.73 million) deaths in 1990 to 1.17 million (95% UI: 0.793–1.62 million) in 2021 [2]. This decline may be attributed to improved treatment options such as oral rehydration solution, zinc supplementation, appropriate use of antibiotics, and vaccination programs targeting rotavirus and other pathogens [3–5]. Other contributing factors include nutrition advancements, such as promoting breastfeeding practices and vitamin A supplementation, as well as improvements in water, sanitation, and hygiene practices [3,4]. Nevertheless, diarrheal diseases are still a leading cause of mortality among children 1–59 months of age, with approximately 340,000 deaths in 2021 [2]. Low-income

regions with limited healthcare access, clean water, and sanitation facilities bear the greatest burden [6], with Africa and Southern Asia accounting for the majority of under-five diarrheal deaths [2]. Furthermore, children in low-income countries experience an average of three episodes of diarrhea per year [2].

The Sustainable Development Goals aim to reduce child mortality to 25 deaths or fewer per 1,000 live births by 2030, with a specific target of less than 1 death per 1,000 live births due to diarrhea [7]. Achieving these targets requires accurate data not only on diarrheal incidence, but also on the specific etiologies and preventable pathways leading to fatal outcomes. Although sentinel studies such as the Global Enteric Multicenter Study (GEMS) [8], the Vaccine Impact on Diarrhea in Africa (VIDA) study [9], the MAL-ED cohort [10], and the Global Pediatric Diarrhea Surveillance Network [11] have provided critical insights into the etiology of moderate-to-severe diarrhea, they primarily focused on morbidity in clinical settings. These studies were not designed to determine the complete sequence of events resulting in death or to attribute specific pathogens to mortality using postmortem diagnostics.

To address this gap, the Child Health and Mortality Prevention Surveillance (CHAMPS) network was launched in 2016 to generate high-resolution data on causes of death in children under five in high-mortality settings in Africa and Southern Asia [12–14]. CHAMPS employs a comprehensive postmortem diagnostic strategy—including minimally invasive tissue sampling (MITS), histopathology, microbial testing, and standardized panel review—to identify and verify pathogen-specific causes of death [15–17]. This approach enables determination of the entire causal chain, from underlying conditions to immediate causes, offering a rare perspective on the syndromic complexity of fatal pediatric diarrhea. Unlike most existing studies, CHAMPS also includes deaths occurring outside healthcare facilities and uses uniform methods across multiple countries, enhancing comparability and generalizability.

Utilizing CHAMPS data, we conducted a descriptive analysis of deceased children aged 1–59 months enrolled in CHAMPS through December 2023 to: (1) quantify the contribution of diarrheal diseases to under-five mortality, (2) examine comorbidities along the causal pathway leading to diarrheal death, and (3) identify etiologic organisms associated with these deaths. This study complements and extends earlier work by offering postmortem-confirmed, pathogen specific data on fatal diarrheal disease. In doing so, it provides new insight into pathogen burden, multisystem interactions, and missed opportunities for prevention, thereby informing diagnostic and treatment requirements and more targeted prevention interventions in resource-limited settings.

## Methods

### Ethics statement

Ethical approval was obtained for CHAMPS from each site's ethical review board, by the Emory University Rollins School of Public Health (Emory Institutional Review Board: 00091706), and by the US Centers for Disease Control and Prevention (CDC). Written informed consent was sought and obtained from primary care givers of all notified child deaths in the CHAMPS catchment areas before data and specimen collection. Each family was provided with the cause of death results of the deceased child.

### CHAMPS cause of death determination

The CHAMPS network operates child mortality surveillance sites in several countries in Africa and south Asia as previously described [18–20]. The network employs a standardized approach that combines specialized clinical and laboratory diagnostics methods, along with clinical record review and verbal autopsy, to define, assign, and code causes of child deaths in children under five [21]. A Determination of Cause of Death (DeCoDe) panel at each site comprises experts from diverse disciplines, including clinical microbiologists, anatomical pathologists, epidemiologists, pediatricians, neonatologists, midwives, obstetricians, and gynecologists. The panels use an extensive range of variables, including laboratory data, microbial assays for a wide variety of pathogens, histopathology results, abstracted clinical records, and findings

from verbal autopsies to determine the underlying, antecedent/comorbid, and immediate causes of death (CoD). The CoD information is recorded on a standardized case report form following the guidelines of the World Health Organization (WHO) death certificate and the International Classification of Diseases, Tenth Revision (ICD-10) [22].

Laboratory assays for potential pathogens include culture of blood and cerebrospinal fluid and multiplex real-time PCR using Taqman Array Cards (TAC) for samples from blood and CSF (same design), lung tissue and naso-/oropharyngeal swabs, and rectal swabs [23]. Enteric pathogens targeted by TAC include norovirus (GI, GII), *Clostridium difficile* (toxin genes tcdA and tcdB, and the pathogenicity locus [paLOC]), *Salmonella* spp., *Entamoeba histolytica, Giardia lamblia, Aeromonas* spp.*, Enterococcus faecium* and *faecalis*, adenovirus (non-typable and 40/41), enterovirus, astrovirus, *Campylobacter coli* and *jejuni*, rotavirus (A, B, C, other), *Yersinia* spp., *Mycobacterium tuberculosis*, sapovirus (I, II, IV, V), *Vibrio cholerae*, cholera toxin, *Cryptosporidium parvum, Ascaris lumbricoides*, and *Trichuris trichuria*. For *Escherichia coli*, the targeted genes and their associated pathotypes are as follows: the *eltB* gene identifies heat-labile enterotoxigenic *E. coli* (LT-ETEC), the *estA* gene identifies heat-stable enterotoxigenic *E. coli* (ST-ETEC), *bfpA* with or without *eae* identifies typical Enteropathogenic *E. coli* (EPEC), *eae* without *bfpA*, *stx1*, or *stx2* identifies atypical EPEC, *aatA* and/or *aaiC* identify Enteroaggregative *E. coli* (EAEC), *eae* with *stx1*, *stx2*, or both, and the absence of *bfpA* identifies enterohemorrhagi*c E. coli* (EHEC), and *ipaH* identifies *Shigella* spp. and enteroinvasive *E. coli* (*Shigella*/EIEC). When the only positive test result was for the *E. coli*/*Shigella uidA* gene in blood, and *Shigella* (defined by *ipaH*) was absent in stool and not identified by culture in blood, the case was classified as having no identified pathogen.

Cycle threshold (Ct) values from qPCR testing were reviewed by CHAMPS site laboratories and DeCoDe panels but were not interpreted using a universal cutoff. Instead, results were assessed qualitatively, incorporating amplification curve shape, fluorescence intensity, assay-specific guidance, and clinical context. [23]. As a general practice, detections with Ct < 35 were considered present, but attribution of causality was left to panel interpretation. While certain assays applied defined thresholds (e.g., Ct > 30 for *E. coli*/*Shigella* due to potential false positives from residual nucleic acid in the PCR mastermix), most pathogen results and corresponding Ct values were interpreted in tandem with full clinical, histopathologic, and epidemiologic context available for each case, to determine pathogen attribution [23].

Aligned with WHO ICD-10 coding, CHAMPS defines diarrhea as liquid or watery or loose stools with increase of 3 episodes of liquid or watery or loose stools above baseline a day for at least 1 day. If the etiologic agent is known, CHAMPS further classifies the diarrhea death based on etiologic agent, for example, gastroenteritis/enteritis due to cholera, enteroinvasive *Escherichia coli* (EIEC) or Shigellosis.

If only a single cause of death is identified, it is listed as the underlying cause. In cases of multiple contributing causes, the panel delineates the causal chain, including the underlying, antecedent, and immediate causes leading to death [15,24]. The underlying cause typically precedes the immediate or antecedent conditions and may have predisposed the child to immediate or antecedent causes leading to death. The immediate cause is the closest event preceding death, while antecedent causes are intermediate between the underlying and immediate causes. In this study, 'malnutrition' was assigned as a cause of death by DeCoDe panels based on clinical records, postmortem anthropometric measurements, and contextual information, and could include underweight, wasting, stunting, or combinations of these deficits, following WHO Child Growth Standards [25,26]. Pathogens associated with the CoD are detected through testing of postmortem brain, liver, and lung tissues, as well as blood, cerebrospinal fluid, nasopharyngeal swab, and rectal swab specimens collected using MITS within 24–36 hours of death and determined to be linked to one of the causes by the DeCoDe panel's review of the all evidence [21]. DeCoDe panels may also attribute a death to a pathogen identified clinical testing conducted before a child's death.

In addition to cause of death determination, DeCoDe panels systematically assessed whether each death was potentially preventable. Preventability was defined based on conditions immediately surrounding the death, including available demographic, clinical, pathological, microbiological, verbal autopsy, and anthropometric information, and did not extend to broader societal, political, or financial factors [24]. For deaths deemed potentially preventable, panels broadly identified

health system gaps that, if addressed, might have prevented the death. In addition, panels categorized these recommendations into predetermined categories of clinical management, health education, and health-seeking behavior.

## CHAMPS surveillance sites

The selection process and characteristics of the CHAMPS sites have been previously described [18]. Briefly, the CHAMPS network includes geographically defined catchment areas in Africa: Bamako in Mali, Kersa and Harar in Ethiopia, Makeni and Bo in Sierra Leone, Manhiça and Quelimane in Mozambique, Siaya and Kisumu in Kenya, and Soweto and Thembelihle in South Africa. In South Asia, CHAMPS operates in Baliakandi and Faridpur in Bangladesh. These sites were selected based on criteria such as child mortality rates of over 50 deaths per 1,000 live births in children under the age of 5 years old at the time of site selection (2015) and willingness to adhere to a standardized multisite protocol and to share data globally in real-time. All child deaths within these catchment areas, whether in the community or healthcare facilities, were eligible for enrollment and were approached for informed consent. CHAMPS countries had introduced rotavirus vaccines into their national immunization programs prior to or during the surveillance period.

## Statistical analyses

Our surveillance analyses focused on deaths in children 1–59 months of age who were enrolled in the CHAMPS network. We included all deaths enrolled from 18 December 2016–31 December 2023 with complete CoD attribution. Data were accessed for analysis on 31 May 2024.

We evaluated the contribution of diarrhea in the causal pathway of childhood deaths, focusing on infants aged 1–11 months and children aged 12–59 months. Newborns and stillbirths were excluded due to distinct clinical presentations, challenges in diagnosing diarrhea in these younger age groups, and potentially differing etiologies and pathophysiological mechanisms. We examined differences in prevalence by sex, CHAMPS site, place of death (community vs. health facility). Additionally, we collected data on rotavirus vaccination status and the presence of co-morbidities among cases with diarrhea. Furthermore, we evaluated pathogens in the causal chain implicated in the diarrheal disease, specifically focusing on co-infections. The analysis was stratified by location of death and site/country. We compared the prevalence and specific pathogen profiles identified from cases with and without diarrhea listed in the causal pathway to death.

Descriptive statistics were calculated, presenting medians with interquartile ranges (IQR) for continuous variables and proportions with 95% confidence intervals (95% CI) for categorical variables. For select variables, differences between infants and children were tested using Kruskall-Wallis or Fisher's exact tests. Anthropometric indicators were derived using the WHO Child Growth Standards [27,28]. WHO Child Growth Standards were used to calculate Z-scores for weight-for-age (WAZ) as a measure of underweight, length-for-age (LAZ) as a measure of stunting (chronic malnutrition) and both weight-for-length (WLZ) and MUAC Z-scores (MUACZ) as a measure of wasting (acute malnutrition) [25,27,28]. Severe wasting was defined as WLZ $< -3$ SD and/or MUACZ $< -3$ SD. Crude and adjusted cause-specific mortality fractions (CSMF) were calculated for each catchment area and site (S1 Methods). All statistical analyses were conducted using R software, version 4.2.3 (R Foundation for Statistical Computing, Vienna, Austria).

## Results

### Deaths among children 1–59 months

Between December 2016 and December 2023, a total of 1517 decedents aged 1–59 months were enrolled across the 7 CHAMPS sites (S1 Fig); with 71.2% of those deaths occurring in health facilities. Median age at death was 11 months (IQR: 4–23), and 52.1% of deaths were among infants. Forty-five percent of decedents were classified as severely underweight, and 28.6% were severely stunted (Table 1). The median age at death varied across sites, ranging from six months in South Africa and Bangladesh to 15 months in Mozambique. Males comprised 54.8% of deaths, with Mali being the only

**Table 1. Characteristics of all MITS enrolled infant and child deaths, CHAMPS Network, 2016–2023 (N = 1517).**

| Characteristics | Total | South Africa | Kenya | Sierra Leone | Mozambique | Mali | Ethiopia | Bangladesh |
|---|---|---|---|---|---|---|---|---|
| | N=1517 | N=339 (22.3%) | N=366 (24.1%) | N=358 (23.6%) | N=258 (17.0%) | N=103 (6.8%) | N=83 (5.5%) | N=10 (0.6%) |
| **Median age at death – months (IQR)** | 11 [4, 23] | 6 [2, 16] | 11 [6, 23] | 14 [6, 25] | 15 [6, 26] | 10 [4, 20] | 14 [5, 25] | 4 [1, 15] |
| **Age group, n (%)** | | | | | | | | |
| Infant (28 days to less than 12 months) | 790 (52.1) | 234 (69.0) | 196 (53.6) | 149 (41.6) | 105 (40.7) | 61 (59.2) | 38 (45.8) | 7 (70.0) |
| Child (12 months to less than 60 Months) | 727 (47.9) | 105 (31.0) | 170 (46.4) | 209 (58.4) | 153 (59.3) | 42 (40.8) | 45 (54.2) | 3 (30.0) |
| **Sex, n (%) (N=1516)** | | | | | | | | |
| Female | 684 (45.1) | 143 (42.2) | 172 (47.0) | 162 (45.3) | 109 (42.2) | 54 (52.9) | 40 (48.2) | 4 (40.0) |
| Male | 832 (54.9) | 196 (57.8) | 194 (53.0) | 196 (54.7) | 149 (57.8) | 48 (47.1) | 43 (51.8) | 6 (60.0) |
| **HIV infection, n (%) (N=1346)** | | | | | | | | |
| Uninfected | 1211 (90.0) | 277 (89.4) | 306 (89.5) | 289 (92.6) | 219 (85.9) | 54 (90.0) | 56 (98.2) | 10 (100.0) |
| Infected | 135 (10.0) | 33 (10.6) | 36 (10.5) | 23 (7.4) | 36 (14.1) | 6 (10.0) | 1 (1.8) | 0 (0.0) |
| **Location of death, n (%)** | | | | | | | | |
| Facility | 1080 (71.2) | 257 (75.8) | 199 (54.4) | 319 (89.1) | 205 (79.5) | 62 (60.2) | 29 (34.9) | 9 (90.0) |
| Community | 437 (28.8) | 82 (24.2) | 167 (45.6) | 39 (10.9) | 53 (20.5) | 41 (39.8) | 54 (65.1) | 1 (10.0) |
| **Median weight (kg) at admission (IQR) (N=836)** | 6.7 [4.4, 9.4] | 4.2 [2.3, 8.5] | 6.8 [4.7, 9.4] | 7.4 [5.5, 10.0] | 6.9 [5.1, 9.4] | 6.3 [4.8, 7.9] | 7.5 [5.4, 10.2] | 4.6 [3.2, 6.8] |
| **Weight-for-age Z-score, n (%) (N=1494)** | | | | | | | | |
| Normal (≥-2SD) | 598 (40.0) | 135 (41.9) | 148 (40.5) | 145 (40.6) | 118 (45.7) | 41 (41.4) | 11 (13.3) | 0 (0.0) |
| Moderate underweight (<-2 SD, -3 SD) | 228 (15.3) | 37 (11.5) | 50 (13.7) | 81 (22.7) | 33 (12.8) | 16 (16.2) | 9 (10.8) | 2 (20.0) |
| Severe underweight (<-3 SD) | 668 (44.7) | 150 (46.6) | 167 (45.8) | 131 (36.7) | 107 (41.5) | 42 (42.4) | 63 (75.9) | 8 (80.0) |
| **Length-for-age Z-score, n (%) (N=1509)** | | | | | | | | |
| Normal (≥-2SD) | 866 (57.4) | 187 (55.7) | 233 (63.7) | 241 (67.5) | 116 (45.0) | 68 (68.7) | 14 (16.9) | 7 (70.0) |
| Moderate stunting (<-2 SD, -3 SD) | 211 (14.0) | 40 (11.9) | 58 (15.8) | 53 (14.8) | 44 (17.1) | 8 (8.1) | 7 (8.4) | 1 (10.0) |
| Severe stunting (< -3 SD) | 432 (28.6) | 109 (32.4) | 75 (20.5) | 63 (17.6) | 98 (38.0) | 23 (23.2) | 62 (74.7) | 2 (20.0) |
| **Weight-for-length Z-score, n (%) (N=1421)** | | | | | | | | |
| Normal (≥ -2SD) | 650 (45.7) | 147 (52.5) | 141 (39.6) | 159 (45.3) | 141 (57.1) | 33 (33.0) | 29 (37.2) | 0 (0.0) |
| Moderate wasting (<-2 SD, -3 SD) | 226 (15.9) | 33 (11.8) | 65 (18.3) | 62 (17.7) | 31 (12.6) | 18 (18.0) | 16 (20.5) | 1 (11.1) |
| Severe wasting (<-3 SD) | 545 (38.4) | 100 (35.7) | 150 (42.1) | 130 (37.0) | 75 (30.4) | 49 (49.0) | 33 (42.3) | 8 (88.9) |
| **Mid-upper arm circumference (cm) Z-score, n (%) (N=1237)** | | | | | | | | |
| Normal (≥ -2SD) | 690 (55.8) | 155 (68.0) | 163 (50.6) | 199 (63.0) | 126 (57.5) | 36 (46.2) | 10 (14.5) | 1 (20.0) |
| Moderate malnutrition (<-2 SD, -3 SD) | 157 (12.7) | 16 (7.0) | 42 (13.0) | 47 (14.9) | 26 (11.9) | 15 (19.2) | 9 (13.0) | 2 (40.0) |
| Severe malnutrition (<-3 SD) | 390 (31.5) | 57 (25.0) | 117 (36.3) | 70 (22.2) | 67 (30.6) | 27 (34.6) | 50 (72.5) | 2 (40.0) |
| **Severe wasting (WLZ or MUACZ<-3 SD), n (%) (N=1458)** | | | | | | | | |
| Yes | 654 (44.9) | 128 (42.7) | 169 (46.7) | 139 (39.3) | 105 (41.7) | 50 (50.0) | 55 (67.9) | 8 (88.9) |
| No | 804 (55.1) | 172 (57.3) | 193 (53.3) | 215 (60.7) | 147 (58.3) | 50 (50.0) | 26 (32.1) | 1 (11.1) |
| **Low birth weight, n (%) (N=508)** | | | | | | | | |

*(Continued)*

Table 1. (Continued)

| Characteristics | Total | South Africa | Kenya | Sierra Leone | Mozambique | Mali | Ethiopia | Bangladesh |
|---|---|---|---|---|---|---|---|---|
| | N=1517 | N=339 (22.3%) | N=366 (24.1%) | N=358 (23.6%) | N=258 (17.0%) | N=103 (6.8%) | N=83 (5.5%) | N=10 (0.6%) |
| Yes | 375 (24.7) | 44 (13.0) | 153 (41.8) | 80 (22.3) | 55 (21.3) | 42 (40.8) | 0 (0.0) | 1 (10.0) |
| No | 133 (8.8) | 62 (18.3) | 27 (7.4) | 14 (3.9) | 16 (6.2) | 12 (11.7) | 0 (0.0) | 2 (20.0) |
| Missing | 1009 (66.5) | 233 (68.7) | 186 (50.8) | 264 (73.7) | 187 (72.5) | 49 (47.6) | 83 (100.0) | 7 (70.0) |
| **Small for gestational age, n (%) (N=380)** | | | | | | | | |
| Yes | 155 (10.2) | 51 (15.0) | 41 (11.2) | 36 (10.1) | 14 (5.4) | 13 (12.6) | 0 (0.0) | 0 (0.0) |
| No | 225 (14.8) | 16 (4.7) | 93 (25.4) | 53 (14.8) | 34 (13.2) | 26 (25.2) | 0 (0.0) | 3 (30.0) |
| Missing | 1137 (75.0) | 272 (80.2) | 232 (63.4) | 269 (75.1) | 210 (81.4) | 64 (62.1) | 83 (100.0) | 7 (70.0) |
| **Median number days between admission and death (IQR) (N=923)** | 1 [0, 6] | 5 [1, 36] | 1 [0, 4] | 1 [0, 3] | 1 [0, 5] | 1 [0, 7] | 3 [2, 13] | 3 [2, 3] |
| **Median hours between death and MITS done (IQR) (N=1515)** | 13 [6, 21] | 24 [17, 41] | 17 [10, 23] | 7 [3, 13] | 12 [5, 19] | 9 [3, 14] | 5 [3, 9] | 2 [1, 3] |
| **Diarrhea not in causal chain (N=1277)** | 1277 (84.2) | 307 (90.6) | 315 (86.1) | 312 (87.2) | 202 (78.3) | 85 (82.5) | 49 (59.0) | 7 (70.0) |
| Antemortem clinical record of diarrhea (%)[a] | 384 (30.1) | 78 (25.4) | 137 (43.5) | 77 (24.7) | 63 (31.2) | 15 (17.6) | 14 (28.6) | 0 (0.0) |
| **Diarrhea was in the causal chain (N=240)** | 240 (15.8) | 32 (9.4) | 51 (13.9) | 46 (12.8) | 56 (21.7) | 18 (17.5) | 34 (41.0) | 3 (30.0) |
| Antemortem clinical record of diarrhea (%)[b] | 212 (89.2) | 28 (87.5) | 51 (100.0) | 43 (93.5) | 50 (89.3) | 12 (66.7) | 27 (79.4) | 3 (100.0) |
| Duration of last diarrhea episode prior to death (median & IQR) (N=127)[b] | 3 [2, 5] | 4 [2, 8] | 3 [2, 3] | 3 [2, 4] | 3 [2, 6] | 6 [5, 7] | 3 [3, 10] | 2 [2, 3] |
| Admitted to ICU (%) | 319 (24.2) | 95 (33.8) | 20 (5.8) | 75 (22.5) | 88 (41.3) | 39 (41.1) | 2 (4.1) | 0 (0.0) |

Weight at admission: South Africa (N = 177), Kenya (N = 322), Sierra Leone (N = 273), Mozambique (N = 162), Mali (N = 35), Ethiopia (N = 8), Bangladesh (N = 4).

Weight-for-age Z-score: South Africa (N = 177), Kenya (N = 322), Sierra Leone (N = 365), Mozambique (N = 357), Mali (N = 99), Ethiopia (N = 83), Bangladesh (N = 10).

Length-for-age Z-score: South Africa (N = 336), Kenya (N = 366), Sierra Leone (N = 357), Mozambique (N = 357), Mali (N = 99), Ethiopia (N = 83), Bangladesh (N = 10).

Days between admission and death: South Africa (N = 220), Kenya (N = 220), Sierra Leone (N = 294), Mozambique (N = 161), Mali (N = 50), Ethiopia (N = 12), Bangladesh (N = 5).

Weight-for-length Z-score: South Africa (N = 280), Kenya (N = 356), Sierra Leone (N = 351), Mozambique (N = 247), Mali (N = 100), Ethiopia (N = 78), Bangladesh (N = 9).

Mid-upper arm circumference Z-score: South Africa (N = 228), Kenya (N = 322), Sierra Leone (N = 316), Mozambique (N = 219), Mali (N = 78), Ethiopia (N = 69), Bangladesh (N = 5).

Duration of last diarrhea episode (diarrhea not in causal chain): South Africa (N = 22), Kenya (N = 76), Sierra Leone (N = 22), Mozambique (N = 29), Mozambique (N = 26), Mali (N = 3), Ethiopia (N = 2), Bangladesh (N = 0).

Duration of last diarrhea episode (diarrhea in causal chain): South Africa (N = 12), Kenya (N = 42), Sierra Leone (N = 22), Mozambique (N = 39), Mali (N = 5), Ethiopia (N = 5), Bangladesh (N = 2).

Days in ICU: South Africa (N = 45), Kenya (N = 18), Sierra Leone (N = 62), Mozambique (N = 71), Mali (N = 26), Ethiopia (N = 2), Bangladesh (N = 0).

Diarrhea not in the causal chain: South Africa (N = 307), Kenya (N = 315), Sierra Leone (N = 312), Mozambique (N = 202), Mali (N = 85), Ethiopia (N = 49), Bangladesh (N = 7).

Diarrhea in the causal chain: South Africa (N = 32), Kenya (N = 51), Sierra Leone (N = 46), Mozambique (N = 56), Mali (N = 18), Ethiopia (N = 34), Bangladesh (N = 3).

a Among deaths without diarrheal disease in the causal chain.

b Among deaths with diarrheal disease in the causal chain.

site where less than half of enrolled deaths were male (47.1%). Among the 1,517 deaths, HIV test results were available for 88.7% (1,346/1,517) of cases, and 10.0% (135/1,346) were HIV-infected. HIV prevalence among tested deaths was highest in Mozambique (14.1%), followed by Kenya (10.5%), South Africa (10.6%), Mali (10.0%), and Sierra Leone (7.4%). Among tested deaths, 1.8% (1/57) in Ethiopia were HIV-infected, and none of the 10 tested deaths in Bangladesh were HIV-exposed or infected. The proportions of severe underweight ranged from 36.7% in Sierra Leone to 75.9% in Ethiopia, and severe stunting ranged from 17.6% in Sierra Leone to 74.7% in Ethiopia. Severe wasting—defined as WLZ or MUACZ < −3 SD—ranged from 39.3% in Sierra Leone to 67.9% in Ethiopia. Eight of ten decedents from Bangladesh were classified as severely underweight and severely wasted.

**Diarrhea as a cause of death**

Of the 1517 decedents evaluated, clinical and verbal autopsy records documented that 593 (39.1%) had diarrhea symptoms prior to death, with the highest percentage observed in Kenya (51.4%, 188/366). When all sites are combined, diarrheal disease was considered to be in the causal chain of death in 240 cases (15.8% of all decedents; Table 2).

**Table 2.** Deaths with diarrheal disease on the causal pathway to death stratified by site, age group, and categorization of whether diarrheal disease was the underlying, antecedent, or immediate cause of death, CHAMPS Network, 2016–2023 (N = 1517).

|  | N | Anywhere in causal pathway[a] | Immediate CoD | Antecedent CoD | Underlying CoD |
|---|---|---|---|---|---|
| Total | 1517 | 240[b] (15.8) | 50 (3.3) | 90 (5.9) | 106 (7.0) |
| 1-11m | 790 | 135 (17.1) | 30 (3.8) | 37 (4.7) | 71 (9.0) |
| 12-59m | 727 | 105 (14.4) | 20 (2.8) | 53 (7.3) | 35 (4.8) |
| South Africa | 339 | 32 (9.4) | 2 (0.6) | 9 (2.7) | 22 (6.5) |
| 1-11m | 234 | 25 (10.7) | 0 (0) | 7 (3.0) | 18 (7.7) |
| 12-59m | 105 | 7 (6.7) | 2 (1.9) | 2 (1.9) | 4 (3.8) |
| Kenya | 366 | 51 (13.9) | 22 (6.0) | 7 (1.9) | 27 (7.4) |
| 1-11m | 196 | 36 (18.4) | 15 (7.7) | 4 (2.0) | 20 (10.2) |
| 12-59m | 170 | 15 (8.8) | 7 (4.1) | 3 (1.8) | 7 (4.1) |
| Sierra Leone | 358 | 46 (12.8) | 18 (5.0) | 14 (3.9) | 14 (3.9) |
| 1-11m | 149 | 25 (16.8) | 9 (6.0) | 4 (2.7) | 12 (8.1) |
| 12-59m | 209 | 21 (10.0) | 9 (4.3) | 10 (4.8) | 2 (1.0) |
| Mozambique | 258 | 56 (21.7) | 4 (1.6) | 16 (6.2) | 36 (14.0) |
| 1-11m | 105 | 28 (26.7) | 3 (2.9) | 9 (8.6) | 16 (15.2) |
| 12-59m | 153 | 28 (18.3) | 1 (0.7) | 7 (4.6) | 20 (13.1) |
| Mali | 103 | 18 (17.5) | 1 (1.0) | 14 (13.6) | 3 (2.9) |
| 1-11m | 61 | 7 (11.5) | 1 (1.6) | 5 (8.2) | 1 (1.6) |
| 12-59m | 42 | 11 (26.2) | 0 (0) | 9 (21.4) | 2 (4.8) |
| Ethiopia | 83 | 34 (41.0) | 3 (3.6) | 28 (33.7) | 3 (3.6) |
| 1-11m | 38 | 12 (31.6) | 2 (5.3) | 7 (18.4) | 3 (7.9) |
| 12-59m | 45 | 22 (48.9) | 1 (2.2) | 21 (46.7) | 0 (0) |
| Bangladesh | 10 | 3 (30.0) | 0 (0) | 2 (20.0) | 1 (10.0) |
| 1-11m | 7 | 2 (28.6) | 0 (0) | 1 (14.3) | 1 (14.3) |
| 12-59m | 3 | 1 (33.3) | 0 (0) | 1 (33.3) | 0 (0) |

[a] For deaths in which only a single cause led to death, that cause is listed as the underlying cause. For deaths in which multiple causes led to the death, the panel determines the causal chain including the underlying, antecedent, and immediate causes leading to death. The underlying cause usually occurred before immediate or antecedent conditions and may have predisposed the child to an immediate cause or co-morbid illnesses that then led to death; the immediate cause was closest to the death and the antecedent causes were in-between the underlying and immediate causes. Each death has only one underlying cause, zero or one immediate cause, and zero to multiple antecedent causes.

Specifically, diarrhea was identified as the immediate cause of death in 50 (3.3%), antecedent cause in 90 (5.9%), and underlying cause in 106 (7.0%) of decedents, with some diarrheal deaths assigned to multiple roles within the causal chain. The proportion of deaths considered to be diarrhea-related varied across the 7 CHAMPS sites, with Ethiopia having the highest proportion at 41.0% (34/83), followed by Mozambique (21.7%, 56/258), Mali (17.5%, 18/103), Kenya (13.9%, 51/366), Sierra Leone (12.8%, 46/358), and South Africa (9.4%, 32/339). Among the few Bangladesh cases, 30% (3/10) were considered associated to diarrhea. Of 437 deaths that occurred in the community, 51 (11.7%) were attributed to diarrheal disease, compared to 189 (17.5%) diarrhea deaths out of 1080 deaths that occurred in health facilities.

There were 240 deaths with diarrheal diseases in the causal chain: six deaths had diarrheal diseases listed as both underlying and immediate causes of death.

Overall, the proportion of deaths attributable to diarrhea was similar between age groups (17.1% in infant vs 14.4% in child deaths, $p = 0.160$) and by sex (15.6% in female vs 16.0% in male deaths, $p = 0.888$). Kenya had a greater proportion of diarrheal deaths among infants (18.4%) than children (8.8%; $p = 0.010$); there were no significant differences between age groups for other sites. Among diarrheal deaths, diarrhea was not usually classified as the immediate cause of death (20.8%, 50/240). Diarrhea was most commonly classified as an underlying cause in South Africa (68.8%, 22/32), Mozambique (64.3%, 36/56), and Kenya (52.9%, 27/51). However, nearly all of these deaths occurred in HIV-negative children, with only one of the 36 cases in Mozambique being HIV-positive. In contrast, diarrhea was frequently categorized as an antecedent cause of death in Ethiopia (82.4%, 28/34) and Mali (77.8%, 14/18) (Table 2).

Among the 240 deaths attributed to diarrhea, 116 (48.7%) were in infants or children who were severely underweight, 41 (33.8%) were severely stunted, and 94 (39.8%) were severely wasted (Table 1). For deaths with diarrheal in the causal chain, proportions that were in children with wasting varied across sites: South Africa (30.0%, 9/30), Kenya (31.4%, 16/51), Sierra Leone (45.6%, 21/46), Mozambique (32.8%, 18/55), Mali (66.7%, 12/18), and Ethiopia (45.5%, 15/33), respectively. Of the 32 children whose deaths were attributed to rotavirus in the causal chain, 8 (25.0%) had a documented record of receiving rotavirus vaccination (S1 Table). Three of these children were aged ≥12 months, raising the possibility of primary vaccine failure or incomplete protection. However, the timing of rotavirus infection relative to vaccination is unknown, and some children may have been infected before completing the vaccine series or before developing adequate immunity.

### Concomitant illnesses in the causal chain of diarrhea-attributed deaths

Among the 240 deaths attributed to diarrhea, 198 (82.5%) had other causes of death identified in the causal pathway. Of the 198, 158 (79.8%) died in a healthcare facility and the median hospital duration was 27 hours (IQR: 10–114 hours). Among these 158 facility deaths, the most common co-morbid causes of death were malnutrition (51.3%), sepsis (46.2%), lower respiratory infections (43.7%), anemia (18.4%), and HIV (11.4%) (Table 3, S2 Fig). Of 42 decedents in whom diarrhea was the only condition in the causal pathway, 31 (73.8%) died in a facility, after a median hospital duration of 17 hours (IQR: 11–43 hours). Notably, half of these deaths occurred within the first 1–2 days of hospitalization, suggesting that diarrhea was acquired in the community and that disease progression was severe and rapid. When diarrheal disease was the underlying cause of death (N = 106), the most frequent immediate or antecedent causes were sepsis (33.0%), lower respiratory infections (25.5%), and other respiratory diseases (8.5%; Table 3). Among these deaths (N = 106), 37 (34.9%) had two total causes of death identified, whereas 27 (25.5%) had three or more causes of death attributed in the causal pathway to death. The most frequent underlying causes of death when diarrheal disease was the immediate or antecedent cause (N = 140) were malnutrition (64.3%) and HIV (13.6%). Table 3 and S3 Fig summarize co-morbidity patterns, including combinations of diarrhea with lower respiratory infections, malnutrition, and sepsis.

When stratifying by site, sepsis was frequently identified as a cause of death alongside diarrhea in most sites, except for Mali (S2 Table). Ethiopia (100%, 34/34), Mali (88.9%, 16/18), Sierra Leone (84.8%, 39/46), Mozambique (83.9%, 47/56), and South Africa (81.2%, 26/32) had the highest proportions of diarrhea deaths with multiple other causes, with

**Table 3. Antecedent/immediate causes of death for infant (1 to <12 months) and child (12–59 months) deaths with diarrheal disease as the underlying cause (N = 106), and underlying causes of death for deaths with diarrheal diseases as the immediate/antecedent cause (N = 140).**

| Cause of death | Diarrheal disease is underlying CoD | | | Diarrheal disease is immediate/antecedent CoD | | |
|---|---|---|---|---|---|---|
| | Total (N = 106) | Infant (N = 71) | Child (N = 35) | Total (N = 140) | Infant (N = 67) | Child (N = 73) |
| Malnutrition | 5 (4.7) | 3 (4.2) | 2 (5.7) | 90 (64.3) | 42 (62.7) | 48 (65.8) |
| Sepsis | 35 (33.0) | 22 (31.0) | 13 (37.1) | 2 (1.4) | 2 (3.0) | 0 (0) |
| Lower respiratory infections | 27 (25.5) | 16 (22.5) | 11 (31.4) | 7 (5.0) | 4 (6.0) | 3 (4.1) |
| Other endocrine, metabolic, blood, and immune disorders | 2 (1.9) | 1 (1.4) | 1 (2.9) | 1 (0.7) | 0 (0) | 1 (1.4) |
| Other neonatal disorders | 1 (0.9) | 1 (1.4) | 0 (0) | 1 (0.7) | 1 (1.5) | 0 (0) |
| Anemias | 5 (4.7) | 2 (2.8) | 3 (8.6) | 0 (0) | 0 (0) | 0 (0) |
| Congenital infection | 0 (0) | 0 (0) | 0 (0) | 1 (1.1) | 0 (0) | 1 (2.0) |
| Heart Diseases | 1 (0.9) | 1 (1.4) | 0 (0) | 0 (0) | 0 (0) | 0 (0) |
| HIV | 0 (0) | 0 (0) | 0 (0) | 19 (13.6) | 9 (13.4) | 10 (13.7) |
| Kidney Disease | 3 (2.8) | 3 (4.2) | 0 (0) | 0 (0) | 0 (0) | 0 (0) |
| Malaria | 0 (0) | 0 (0) | 0 (0) | 4 (2.9) | 1 (1.5) | 3 (4.1) |
| Meningitis/Encephalitis | 2 (1.9) | 1 (1.4) | 1 (2.9) | 0 (0) | 0 (0) | 0 (0) |
| Neonatal preterm birth complications | 0 (0) | 0 (0) | 0 (0) | 1 (0.7) | 1 (1.5) | 0 (0) |
| Other | 2 (1.9) | 2 (2.8) | 0 (0) | 2 (1.4) | 1 (1.5) | 1 (1.4) |
| Other disorders of fluid, electrolyte and acid-base balance | 6 (5.7) | 6 (8.5) | 0 (0) | 0 (0) | 0 (0) | 0 (0) |
| Other infections | 3 (2.8) | 0 (0) | 3 (8.6) | 0 (0) | 0 (0) | 0 (0) |
| Other neurological disorders | 4 (3.8) | 3 (4.2) | 1 (2.9) | 0 (0) | 0 (0) | 0 (0) |
| Other respiratory disease | 9 (8.5) | 6 (8.5) | 3 (8.6) | 0 (0) | 0 (0) | 0 (0) |
| Paralytic ileus and intestinal obstruction | 1 (0.9) | 1 (1.4) | 0 (0) | 0 (0) | 0 (0) | 0 (0) |
| Congenital birth defects | 0 (0) | 0 (0) | 0 (0) | 4 (2.9) | 1 (1.5) | 3 (4.1) |
| Measles | 0 (0) | 0 (0) | 0 (0) | 3 (2.1) | 2 (3.0) | 1 (1.4) |

There were 240 deaths with diarrheal diseases in the causal chain: six deaths had diarrheal diseases listed as both underlying and immediate causes of death.

sepsis and lower respiratory infections being the most common additional causes. HIV prevalence among diarrhea deaths was highest in South Africa (15.6%, 5/32), Mozambique (10.7%, 6/56), and Kenya (7.8%, 4/51). Among deaths where diarrhea was classified as an immediate or antecedent cause, sepsis was often identified as the underlying cause (35 cases), whereas among deaths where diarrhea was the underlying cause, sepsis was infrequently listed as an immediate or antecedent condition (2 cases) (S2 Fig). These findings suggest that in many cases, diarrhea may have developed during or after a systemic illness, rather than initiating it.

### Enteric pathogens associated with diarrheal deaths

A wide range of pathogens was detected from rectal swabs in both diarrhea and non-diarrhea cases (Fig 1); however, the majority of detected organisms were not ultimately determined to be in the causal chain leading to death. Of the 240 deaths attributed to diarrhea, 227 (94.6%) had one or more pathogens known to cause diarrhea detected via TAC PCR testing of rectal swab specimens, though these detections were not necessarily linked to the cause of death (median: 3 pathogens detected, IQR: 2–5, range: 1–11) (Fig 1). Among the 119 diarrheal deaths not attributed to a specific pathogen, 50 (42.0%) had malnutrition identified in the causal pathway. Across these 119 cases, 107 (89.9%) had at least one diarrhea-associated pathogen detected via TAC PCR, though no pathogen was ultimately attributed as the cause of death. The median number of pathogens detected in these cases was 3 (IQR: 2–4; range: 1–7). Among the 121 diarrheal

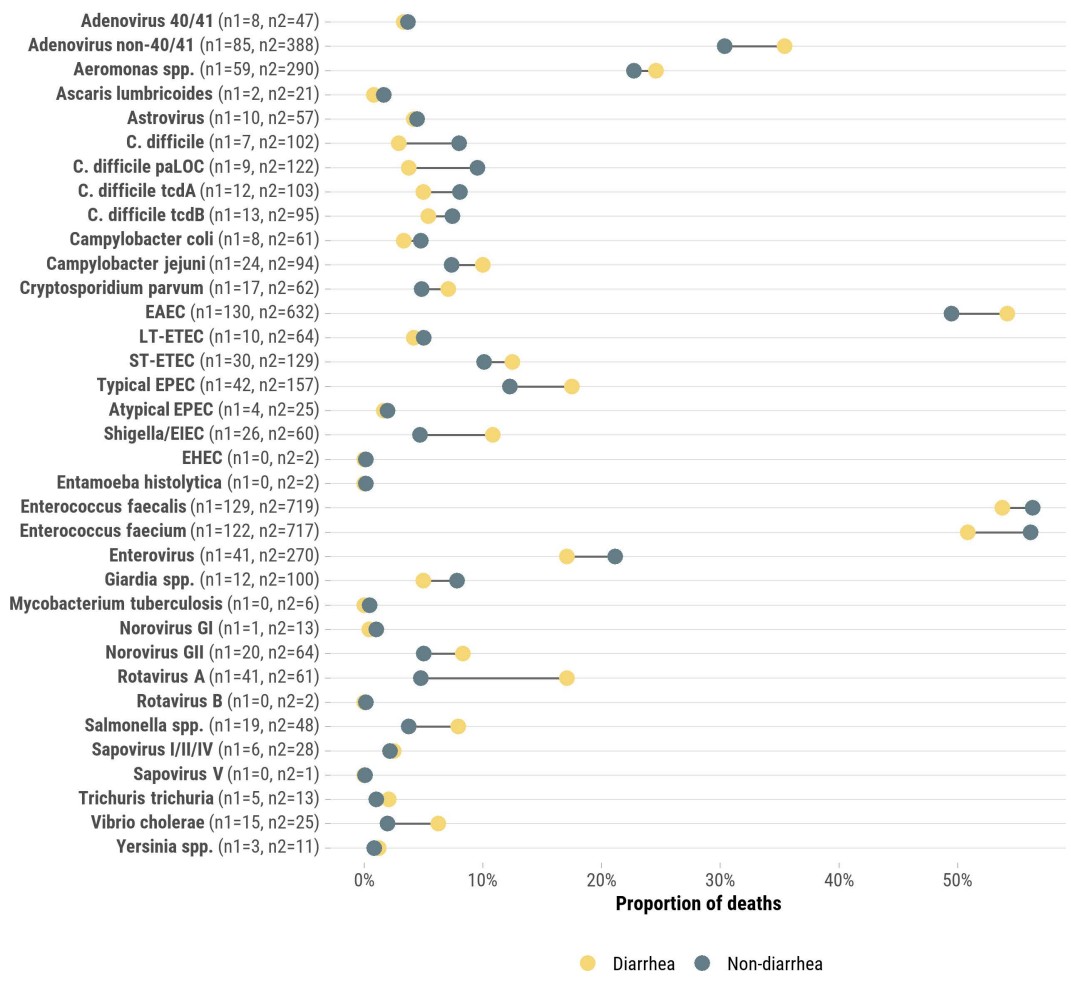

**Fig 1. Proportion of infant and child deaths (N =1517) for which specific pathogens were identified in stool stratified by whether the death was attributed to diarrhea (N1 = 240, yellow) or other non-diarrhea (N2 = 1277, green) causes, CHAMPS Network, 2016–2023.** Pathogens detected from rectal swabs in both diarrhea and non-diarrhea cases via TAC PCR. Detection does not necessarily indicate causation in the fatal event.

deaths attributed to a specific pathogen, the median number of pathogens detected was 4 (IQR: 3–6, range: 1–11). For comparison, 1190 of 1277 (93.2%) deaths that did not have diarrhea anywhere in the causal chain also had diarrhea-associated pathogens identified by TAC on rectal swabs (median 3 pathogens detected, IQR: 2–4, range: 1–11). While multiple pathogens were often detected, DeCoDe panels relied on clinical data, histopathology, and overall disease presentation—including the presence of diarrhea—to determine the most likely cause of death in each case. Adenovirus, *C. difficile paLOC,* EAEC, typical EPEC, *Shigella*/EIEC, rotavirus A, and *V. cholerae* was more likely to be detected in rectal swabs from diarrhea deaths compared to non-diarrhea deaths (Fig 1).

Of the 121 diarrhea deaths in which at least 1 pathogen was attributed by DeCoDe, the number of pathogens attributed to diarrhea per death ranged from 1 to 3 (median 1; Figs 2, 3). For the 121 deaths attributed to a pathogen, the most frequent were EAEC (34.7%, N=42), rotavirus A or non-typable (26.4%, N=32), adenovirus 40/41 or non-40/41 (24.0%, N=29), typical EPEC (15.7%, N=19), *Shigella*/EIEC (14.0%, N=17), and ST-ETEC (12.4%) (Tables 4, S3). Stratifying by age group, adenovirus 40/41 or non-40/41 (17.1% vs. 8.1%; *p*=0.045) was more frequently observed in child deaths than in infant deaths. To assess whether EAEC attribution might be confounded by nutritional status, we compared pathogen

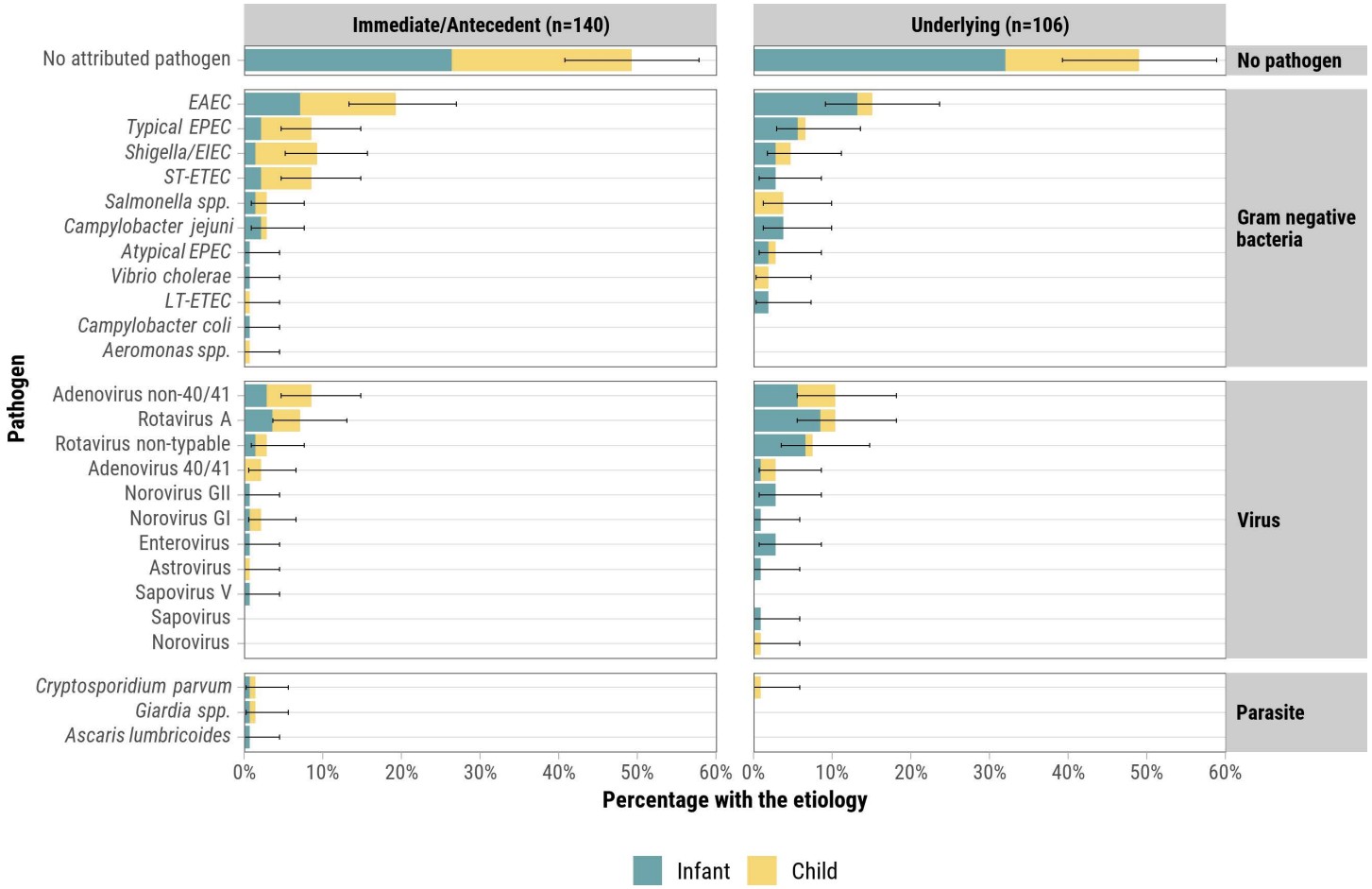

**Fig 2. Pathogens attributed to deaths that had diarrheal disease as immediate or antecedent cause of death (left) or as underlying cause of death (right), CHAMPS Network, 2016–2023.** There were 240 deaths with diarrheal diseases in the causal chain: six deaths had diarrheal diseases listed as both underlying and immediate causes of death. Pathogens without a sub-type were non-typable for example norovirus vs norovirus GII. Pathogens were often found in combination (Fig 5).

distributions between diarrhea deaths with and without malnutrition listed either in the causal chain or as another significant condition (S4 Table). EAEC was more common among malnourished cases (20.7% vs. 13.3%, $p = 0.185$), but remained present in deaths without malnutrition, suggesting a potential role beyond nutritional comorbidity. To further explore differences in attributed pathogens by nutritional and HIV status, we examined pathogen distributions stratified by malnutrition, HIV infection, and absence of both conditions (S5 Table).

The pathogen profile remained similar when considering diarrhea as either an underlying cause or immediate/antecedent cause (Fig 2). Compared with community deaths, a greater proportion of facility deaths were attributed to non-40/41 adenovirus (10.6% vs. 5.6%) and rotavirus A or non-typable (15.3% vs. 5.9%) though these differences were not statistically significant. Fewer facility deaths were attributed to ST-ETEC (2.6% vs. 19.6%) and *Shigella*/EIEC (4.8% vs. 15.7%) (*p*-values≤0.017) (Fig 4). *Cryptosporidium parvum, Giardia spp.,* and *Clostridium difficile* were implicated in three, two, and zero deaths, respectively.

Sixty (25.0%) of the 240 infant and child deaths with diarrheal diseases in the causal chain were found to have co-infections with multiple diarrhea-related pathogens (Fig 5A). Among these, 38 children had two pathogens, whilst 22

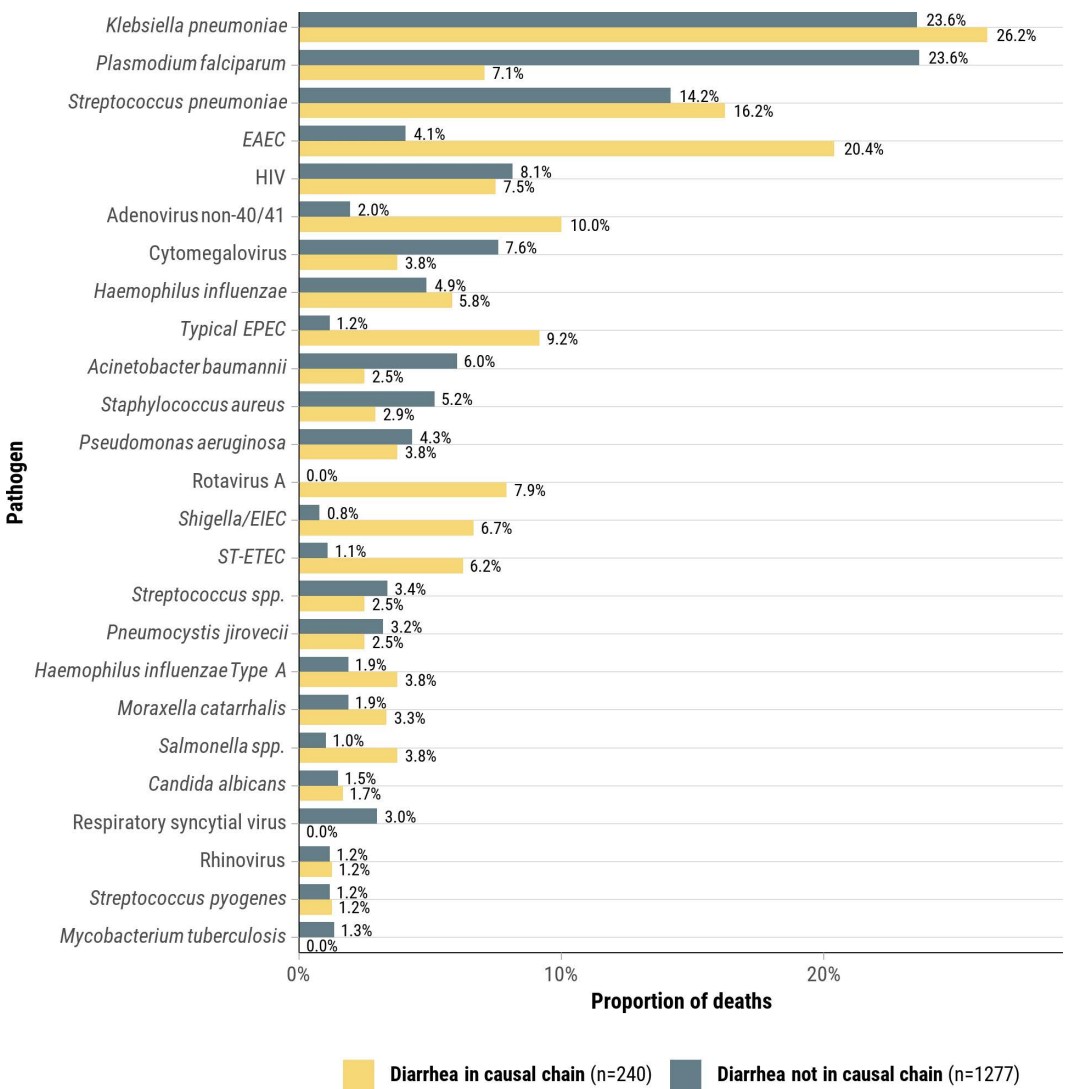

**Fig 3. Pathogens anywhere in the causal chain for deaths due to diarrheal disease compared to pathogens anywhere in the causal chain for deaths without diarrheal disease.** This plot is restricted to the 25 most frequent pathogens, CHAMPS Network, 2016–2023.

**Table 4. Pathotypes of *Escherichia coli* and *Shigella* with associated gene targets and frequencies for 54 diarrheal deaths attributed to *Escherichia coli* or *Shigella*, CHAMPS Network, 2016–2023.**

| Pathotype | Full name | Gene target(s) | n (%) |
|---|---|---|---|
| LT-ETEC | Heat-labile Enterotoxigenic *Escherichia coli* | *eltB* only | 3 (5.6) |
| ST-ETEC | Heat-stable Enterotoxigenic *Escherichia coli* | *estA* only, or both *eltB* and *estA* | 15 (27.8) |
| Typical EPEC | Typical Enteropathogenic Escherichia coli | *bfpA* with or without *eae* | 19 (35.2) |
| Atypical EPEC | Atypical Enteropathogenic Escherichia coli | *eae* without either *bfpA*, *stx1*, or *stx2* | 4 (7.4) |
| EAEC | Enteroaggregative *Escherichia coli* | *aatA*, *aaiC*, or both | 42 (77.8) |
| EHEC | Enterohemorrhagic *Escherichia coli* | *eae* with *stx1*, *stx2*, or both, and the absence of *bfpA* | 0 (0.0) |
| *Shigella*/EIEC | *Shigella*/Enteroinvasive *Escherichia coli* | Detected via *ipaH* | 17 (31.5) |

PLOS Global Public Health

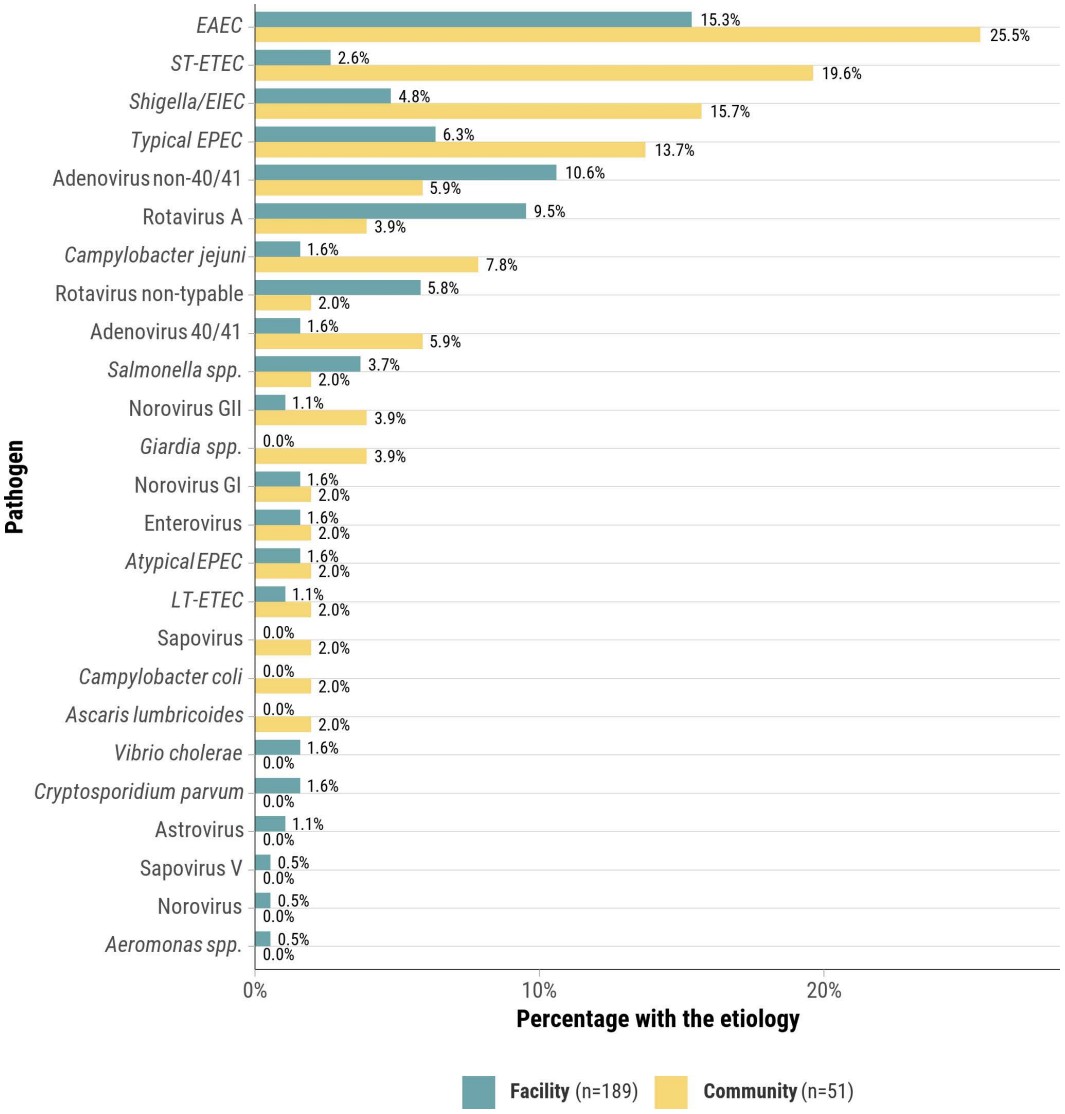

**Fig 4. Comparison of pathogens attributed to health facility diarrheal deaths to those attributed to community diarrheal deaths, CHAMPS Network, 2016–2023.** This figure shows the proportion of pathogens attributed to diarrheal deaths that occurred in the health facility compared to pathogens attributed to diarrheal deaths that occurred in the community. For example, 19.6% of diarrheal deaths that occurred in the health facility were due to ST-ETEC whilst 2.6% of diarrheal deaths that occurred in the community were due to ST-ETEC.

had 3 or more pathogens. Combinations found included bacteria-bacteria (n = 38), bacteria-virus (n = 22), and virus-virus (n = 8). *E. coli* pathotypes were frequently identified together such as EAEC with typical EPEC (n = 14), *Shigella*/EIEC, and ST-ETEC (n = 11) (Fig 5B). Even though diarrhea-related pathogens were isolated in some non-diarrheal deaths, the prevalence of diarrheal pathogens in non-diarrheal deaths was significantly less (S4 Fig).

**Diarrheal disease mortality fractions**

Adjusted cause-specific mortality fractions for diarrheal diseases were highest for sites in Ethiopia (17.6, 90% Credible Interval [CrI]: 15.9, 19.4), followed by Mozambique (8.9, 95% CrI: 7.6, 10.3) and Kenya (5.7, 95% CrI: 4.9, 6.6) (S6 and S7 Tables).

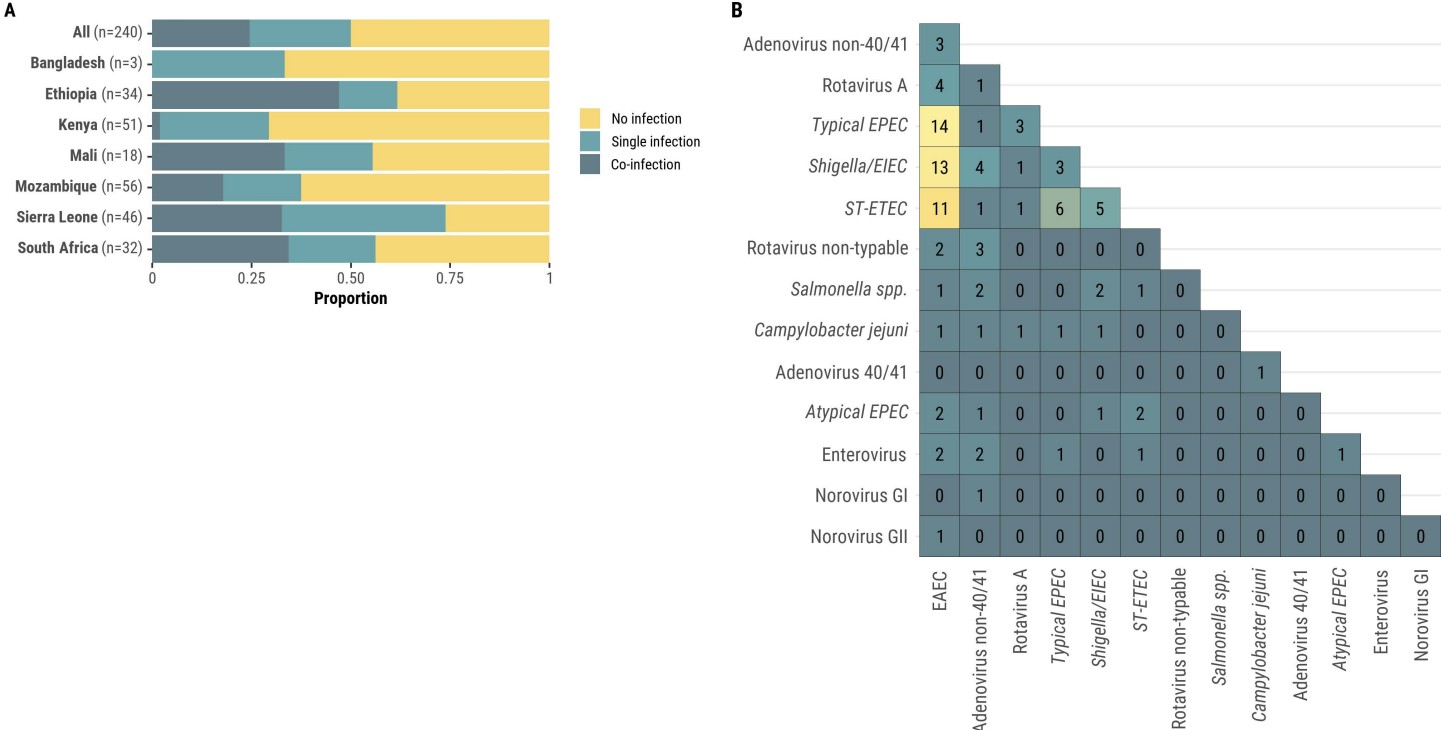

**Fig 5. (A) Proportion of infant and child diarrheal deaths for which the diarrheal disease was attributed to a single infection, co-infection, or no identified infection (N = 240).** (B) Frequency of co-infections associated with death from diarrheal diseases among 240 diarrheal deaths in infants and children (N = 59), CHAMPS Network, 2016–2023. 5B represents pathogen co-infections from 59 diarrheal deaths where each pathogen was present in at least 3 diarrheal deaths. Pathogens implicated in one or two diarrheal deaths, not shown in 5B, include *Aeromonas spp.*, *Ascaris lumbricoides*, astrovirus, *Campylobacter coli*, *Giardia spp.*, norovirus, sapovirus, and sapovirus V.

## Preventability

The DeCoDe panel deemed 91.7% (220/240) of diarrhea deaths as potentially preventable based on the presence of modifiable factors surrounding the death, such as delayed care-seeking, missed clinical opportunities, or inadequate treatment. For 97.3% (214/220) of these preventable deaths, the panels identified specific health system gaps that could have altered the outcome. The most common recommendations were improved clinical management and quality of care for diarrheal illness (69.2%), enhanced health education for caregivers (50.9%), improved health-seeking behavior (44.4%), and greater access to nutritional support (36.0%; S8 Table).

## Discussion

In CHAMPS surveillance sites, diarrheal diseases were attributed to 15.8% of deaths in infants and children, with a marked variation across sites (Ethiopia: 41.0%, South Africa: 9.4%), highlighting regional disparities and confirming that diarrhea remains among the leading causes of infant and child deaths. Diarrhea was more commonly an underlying cause than an immediate cause of death. Improving quality of care for those with diarrhea is critical, with most diarrhea deaths (91.7%) deemed potentially preventable by DeCoDe panels through identification of modifiable health system gaps, including improved clinical management, early detection and management of HIV and malnutrition, health education, and better health-seeking behavior [24,29].

This study addresses a critical knowledge gap by characterizing the etiology and syndromic complexity of fatal diarrhea using postmortem MITS diagnostics. For example, landmark studies such as GEMS and VIDA provided key

insights into the relationship between moderate-to-severe diarrhea, linear growth faltering, and death. GEMS demonstrated that the magnitude of linear growth faltering at enrolment among children with MSD was inversely associated with risk of dying in all age groups. GEMS also characterized the bilateral relationship between diarrhea and linear growth faltering, also apparent in CHAMPS, showing that MSD cases were significantly more likely than their matched controls to experience declining growth velocity. Moreover, they had an 8.5-fold higher risk of dying during the 2–3 months after their illness. However, 67% of deaths in GEMS occurred more than one week after study enrollment, and the immediate cause of death was not clearly elucidated. Another sentinel study, MAL-ED, evaluated diarrheal diseases in a community cohort in which deaths were rare and the primary focus was diarrheal morbidity. CHAMPS provides pathogen-specific attribution for deaths using a combination of MITS, histopathology, and expert panel review. This allows for closer examination of causal, not just associative, classification of pathogens and co-morbid conditions. Importantly, CHAMPS includes both facility and community deaths. CHAMPS identified a similar panel of enteropathogens as was seen in other studies, particularly rotavirus, tEPEC, *Shigella*/EIEC, and ST-ETEC. CHAMPS findings illustrated the well-documented foreboding interplay between diarrhea and malnutrition, with or without HIV, and identified the co-morbidities like sepsis and pneumonia frequently observed as the terminal event among children with diarrhea and malnutrition.

Findings from GEMS, VIDA, and MAL-ED studies have underscored the importance of understanding local and regional pathogen distributions to effectively prioritize interventions [8–10]. These studies demonstrated that pathogens such as rotavirus, ST-ETEC, tEPEC, *Shigella spp.* and *Cryptosporidium* are among the leading contributors to diarrheal morbidity and mortality, findings that resonate with our study's results. In particular, molecular diagnostic techniques used in these studies have enhanced pathogen identification and provide a model for the CHAMPS network's approach to determining diarrheal etiology. However, the sensitivity of these tests may also detect asymptomatic colonization in which case controls without diarrhea or the use of cycle threshold cut-offs may assist in assigning an etiologic attribution. Going forward, incorporating pathogen quantity thresholds, validation against clinical and histopathologic evidence, and expanding the use of control groups may help refine attribution of causality. These approaches could improve interpretation of molecular diagnostic results and enhance precision in identifying true etiologic pathogens.

Sepsis and lower respiratory infections were the most common immediate or antecedent causes of diarrheal deaths among children under five, corroborating existing evidence of the synergistic relationship between diarrheal disease and other infectious conditions, particularly those affecting the respiratory and immune systems [30,31]. Malnutrition and HIV were the most frequent underlying causes when diarrheal diseases was the immediate or antecedent cause, showing how diarrhea can complicate these conditions and lead to fatality. Malnutrition was notably prevalent among these deaths, highlighting its critical role in fatal diarrhea. This aligns with findings from GEMS, which demonstrated that lower HAZ scores were strongly associated with increased risk of mortality in children with moderate-to-severe diarrhea across all age groups ($p < 0.001$ for 0-11m, 12-23m, 24-59m) [32]. Conversely, diarrheal diseases not only result from malnutrition but also contribute to it—through mechanisms such as anorexia, intestinal injury and dysfunction, and catabolism [33]—thereby weakening immune defenses and establishing a vicious cycle in which malnutrition and diarrhea exacerbate each other [25,30,34–36]. The frequent co-occurrence of diarrhea, malnutrition, sepsis, and pneumonia in CHAMPS deaths may reflect an interrelated sequence of vulnerabilities rather than isolated conditions [17,25]. Preventing diarrhea may help interrupt this pathway and reduce mortality in vulnerable children [37]. A similar pattern has been noted in prior studies, including prognostic models identifying children at high risk of death following presentation for diarrheal care [37]. These observations are consistent with recent CHAMPS network findings, which highlight the dominant role of malnutrition and infections in the causal chain of child deaths and emphasize that most deaths are potentially preventable [19]. This underscores the need to move beyond isolated clinical interventions and adopt a broader structural approach to child survival, as advocated by Subramanian [38]. These observations reinforce the importance of addressing malnutrition and immunodeficiencies in the prevention of fatal diarrheal diseases. In particular, they highlight the need for routine screening

for malnutrition and HIV and longitudinal follow-up for children presenting with diarrheal diseases, coupled with early targeted interventions such as nutritional rehabilitation, antiretroviral therapy initiation, and prophylactic treatment of common infections.

Our study identified EAEC, tEPEC, *Shigella*/EIEC, ST-ETEC, rotavirus, and adenovirus non-40/41 as the most frequently identified diarrheal pathogens among children dying from diarrhea. These findings highlight that rotavirus continues to be as a leading cause of severe diarrhea and associated deaths in young children worldwide despite the broad introduction of vaccine [39]. However, differences exist in the relative burden of specific pathogens across sites, underscoring the importance of local epidemiologic data in understanding site-specific diarrheal disease etiology [2]. It is notable that most cases with rotavirus in the causal pathway to death were unvaccinated children or children whose vaccination status could not be verified, although incomplete vaccination records represent a limitation of the available data. Although the proportion of diarrhea deaths attributed to adenovirus was lower than observed in studies like GEMS [8], this may reflect differences in case definitions, secular trends in disease incidence, the timing of sampling, or the rigorous DeCoDe panel process, which requires integration of clinical, histopathologic, and laboratory findings to determine causal attribution. While some subjectivity is inherent in expert panel adjudication, the CHAMPS DeCoDe approach strengthens causal inference by systematically reviewing comprehensive clinical and postmortem data for each case.

Although EAEC was more frequently attributed among malnourished children (20.7% vs. 13.3%), its consistent presence across both groups suggests a potentially underrecognized role in diarrhea-related mortality. One possible explanation is that persistent EAEC infection may contribute to chronic enteric dysfunction and heightened vulnerability to severe illness, especially in settings of repeated enteric exposures and limited access to care [40]. Alternatively, EAEC may be acting synergistically with other pathogens or comorbidities in ways that are not fully captured by current attribution frameworks. In addition, we cannot exclude the possibility of attribution bias, whereby panelists may have been more likely to attribute the episode to EAEC in malnourished children than in those without malnutrition. These findings underscore the need for further investigation into the pathogenic potential of EAEC and its relevance for intervention strategies targeting high-risk children.

The substantial proportion of diarrheal deaths (49.6%) for which a specific pathogen could not be identified despite thorough laboratory testing could be ascribed to the timing of death; children often die after several days of diarrhea, during which the pathogen causing the illness might no longer be detectable due to the immune response or antibiotic treatment. Non-infectious causes could also contribute, like malnutrition, environmental enteropathy or HIV. Notably, many diarrhea deaths without an attributed pathogen still had one or more enteric pathogens detected by PCR, raising the possibility that attribution may have been deprioritized in cases with competing co-morbidities.

The higher proportion of diarrheal deaths occurring in health facilities may reflect better clinical documentation and diagnostic capacity by healthcare professionals, whereas in some settings, caregivers may perceive diarrhea as a common childhood illness and not seek care, leading to underreporting in community deaths. This highlights the potential for misclassification in community-based mortality estimates and suggests that strengthening community awareness and access to care could help capture the true burden of diarrheal disease-related mortality. However, we cannot definitively determine whether more diarrheal deaths occurred in the community without reaching care, as CHAMPS relies on timely death notifications and consent for postmortem enrollment, which may underrepresent community deaths.

Diarrheal disease is a significant contributor to child mortality in low-income countries [24]. Analysis by the DeCoDe panels revealed that most diarrhea deaths (91.7%) were potentially preventable. The association between HIV, malnutrition and diarrheal deaths calls for community and health facility interventions to increase screening and management of all three conditions. Improved clinical management and quality of care, along with better health education and health-seeking behaviors, were identified as the most common areas for intervention. Indeed, CHAMPS in collaboration with local healthcare facilities has implemented simple rehydration units in health facilities where CHAMPS operates resulting in improved

recovery rates for children presenting with diarrhea. Additionally, implementation and scale up of established preventive strategies such as increased access to safe water and sanitation, handwashing with soap, exclusive breastfeeding for the first six months, and rotavirus vaccination have potential to reduce deaths attributable to diarrheal diseases [24,39,41,42].

Limitations of the CHAMPS methodology have been previously described [18,21,43,44]. A major limitation is the lack of control children in CHAMPS which constrained the ability to attribute the enteric pathogens that were identified as the cause of diarrhea, as was possible in other studies such as GEMS and VIDA [8,9]. In CHAMPS, attribution was a clinical assessment made by the adjudication panels. An additional limitation specific to diarrheal deaths is that MITS does not directly confirm diarrhea as a cause of death, but rather identifies associated pathogens—unlike other syndromes (e.g., pneumonia or meningitis) where histopathology or pathogen detection in target tissues adds diagnostic clarity. Other limitations include the inability to capture all deaths within catchment areas, differences in population characteristics between sites, and overrepresentation of healthcare facility deaths. Because CHAMPS relies on timely notification and consent for postmortem procedures, deaths occurring in health facilities—where identification and enrollment are more feasible—are disproportionately represented. As a result, deaths that occur in the community or after hospital discharge may be under-represented, limiting the generalizability of our findings to all child deaths in these settings. Additionally, CHAMPS sites were selected based on factors such as high under-five mortality and existing partnerships with local health authorities, which may further limit broader applicability. The small number of deaths from Bangladesh (N = 10) also restricts interpretability for that site. The high prevalence of underweight observed in diarrhea-related deaths may be partially attributed to dehydration, which can transiently lower weight but not height. Incomplete antemortem vaccination records for rotavirus and other diarrheal pathogens limited our ability to assess the full contribution of vaccination status to the observed mortality rates. Additionally, as diarrhea can be challenging to document retrospectively, some cases may not have had diarrhea recorded despite pathogen detection, highlighting the complexities of attributing enteric pathogens to fatal diarrhea cases.

While our study aimed to quantify the impact of diarrheal diseases on child mortality and explore co-morbidities and co-infections, our analysis focused on deaths where diarrhea was identified as a cause of death. This design will not capture the full spectrum of diarrheal diseases, as cases with milder symptoms, recurrent diarrhea that led to malnutrition, or those who were recovering from diarrhea but had subsequent complications that led to death may not have been included. Incomplete clinical information, timing of sample collection, and potential for post-mortem bacterial overgrowth can all influence the interpretation of these findings. That notwithstanding, our current study contributes to better understanding of the burden and underlying factors associated with fatal diarrheal diseases among children and proposes data driven interventions that can be implemented to reduce such deaths in resource limited settings with high child mortality.

These findings support several actionable priorities for policymakers and public health authorities across the health care pathway. First, at the point of facility-based care, the high proportion of preventable deaths underscores the need to strengthen clinical management of diarrheal illness, including timely rehydration, antimicrobial stewardship, and integration of nutritional support. Second, for children presenting with co-morbid conditions such as malnutrition, HIV, and sepsis, integrated diagnosis and care approaches are critical. Third, upstream prevention efforts should focus on increasing rotavirus vaccine coverage and improving immunization record completeness. Fourth, increasing diarrhea awareness and strengthening community-based case detection through engagement of community health workers is critical to promote early recognition and care-seeking.

These findings are applicable to other regions with similar health systems infrastructure, social, economic, and cultural contexts, and they provide evidence to support the design of surveillance-informed public health programs that align with country-level and global targets, including the SDG goal of reducing child mortality to fewer than 25 deaths per 1,000 live births by 2030. Embedding these strategies into existing child health platforms can accelerate progress toward ending preventable child deaths due to diarrhea.

## Supporting information

**S1 Fig. Flowchart of enrolled under-five infant and child deaths from CHAMPS sites between December 2016 – December 2023, that had minimally invasive tissue samples (MITS) and consent only for verbal autopsy and clinical abstraction (Non-MITS) and included in the analysis.**
(PNG)

**S2 Fig. Underlying causes of death when diarrheal disease was immediate or antecedent cause (left) and immediate or antecedent causes of death when diarrheal disease was underlying cause (right), CHAMPS Network, 2016–2023.** There were 240 deaths with diarrheal diseases in the causal chain: six deaths had diarrheal diseases listed as both underlying and immediate causes of death.
(JPG)

**S3 Fig. Common comorbidity patterns in diarrhea-attributed child deaths, CHAMPS Network, 2016–2023.**
(JPG)

**S4 Fig. Diarrheal pathogens in deaths attributable to diarrheal diseases (yellow) and diarrheal pathogens in deaths not attributable to diarrheal diseases (green), CHAMPS Network, 2016–2023.**
(JPG)

**S1 Table. Rotavirus vaccination status for infant and child deaths, CHAMPS Network, 2016–2023 (N = 1517).**
(DOCX)

**S2 Table. Other causes of death for infants (1 to <12 months) and children (12–59 months) in the causal chain for deaths with diarrheal disease as the underlying cause (N = 106) by site, CHAMPS Network, 2016–2023.**
(DOCX)

**S3 Table. Pathogens attributed to diarrheal disease in the causal chain by age group, CHAMPS Network, 2016–2023.**
(DOCX)

**S4 Table. Pathogens attributed to diarrheal disease in the causal chain for infant and child deaths with and without malnutrition in causal chain or as other significant condition, CHAMPS Network, 2016–2023.**
(DOCX)

**S5 Table. Pathogens attributed to diarrheal disease among infant and child deaths, stratified by nutritional and HIV status, CHAMPS Network, 2016–2023.**
(DOCX)

**S6 Table. Crude and adjusted total under-five mortality fractions and 90% Bayesian credible intervals due to diarrheal diseases at all sites and catchments within the CHAMPS Network.**
(DOCX)

**S7 Table. Crude and adjusted total under-five mortality rates and 90% Bayesian credible intervals due to diarrheal diseases at all sites and catchments within the CHAMPS Network.** S5 Table. Minimally invasive tissue sampling (MITS) procedures that were deemed essential for determining causes of death by age group for deaths with diarrheal diseases in the causal chain, CHAMPS Network, 2016–2023.
(DOCX)

**S8 Table. Expert (DeCoDe) panel recommendations for preventing diarrheal diseases deaths, CHAMPS Network, 2016–2023.**
(DOCX)

**S1 Methods. Cause-Specific Mortality Fractions.**
(DOCX)

## Acknowledgments

The CHAMPS Consortium, non-author contributors: Fatima Solomon, MD; Gillian Sorour, MD; Hennie Lombaard, MD; Jeannette Wadula, MD; Karen Petersen, MD; Martin Hale, MD; Nelesh P. Govender, MD; Peter J. Swart, MD; Sithembiso Velaphi, PhD; Richard Chawana, PhD; Yasmin Adam, MD; Amy Wise, MSc; Nellie Myburgh, PhD. Sanwarul Bari, MD; Shahana Parveen, MSS; Mohammed Kamal, PhD; A.S.M. Nawshad Uddin Ahmed, FCPS; Mahbubul Hoque, FCPS; Saria Tasnim, FCPS; Ferdousi Islam, FCPS; Farida Ariuman, FCPS; Mohammad Mosiur Rahman, MD; Ferdousi Begum, MD; Mustafizur Rahman, PhD; Dilruba Ahmed, PhD; Meerjady Sabrina Flora, PhD; Tahmina Shirin, PhD; Mahbubur Rahman, MPH; Joseph Oundo, PhD; Alexander M. Ibrahim, MD; Fikremelekot Temesgen, MD; Tadesse Gure, MD; Addisu Alemu, MD; Melisachew Mulatu Yeshi, MD; Mahlet Abayneh Gizaw, MD; Stian MS Orlien, PhD; Solomon Ali, PhD; Kitiezo Aggrey Igunza, BSc; Peter Otieno, MA; Peter Nyamthimba Onyango, MA; Janet Agaya, MPH; Richard Oliech, Diploma in lab sciences; Joyce Akinyi Were, MSc; Dickson Gethi, BSc; George Aol, MA; Thomas Misore, MA; Harun Owuor, MSc; Christopher Muga, BSc; Christine Ochola, Diploma in Clinical Medicine & Surgery; Sharon M. Tennant, PhD; Carol L. Greene, MD; J. Kristie Johnson, PhD; Brigitte Gaume, PhD; Rima Koka, MD; Karen D. Fairchild, MD; Diakaridia Kone, MD; Diakaridia Sidibe, MD; Doh Sanogo, MD; Uma U. Onwuchekwa, MSc; Nana Kourouma, MD, PhD; Seydou Sissoko, MD; Cheick Bougadari Traore, MD; Jane Juma, MS, HND in Biotechnology; Kounandji Diarra, MSc; Awa Traore, MSc; Tiéman Diarra, PhD; Kiranpreet Chawla, MD; Tacilta Nhampossa; Zara Manhique; Sibone Mocumbi; Clara Menéndez; Khátia Munguambe; Ariel Nhacolo; Maria Maixenchs; Andrew Moseray, MSc; Fatmata Bintu Tarawally, MSc; Martin Seppeh, BSc; Ronald Mash, DrPH; Julius Ojulong, MD; Babatunde Duduyemi, FMCPath; James Bunn, MD; Alim Swaray-Deen, FWACS - Ob/Gyn; Joseph Bangura, MPH; Amara Jambai, MSc; Margaret Mannah, MPH; Okokon Ita, FMCPath; Sulaiman Sannoh, MD; Princewill Nwajiobi, FMCPath; Dickens Kowuor, MSc; Oluseyi Balogun, MHM; Solomon Samura, BSc; Samuel Pratt, MPH; Francis Moses, Master of Medicine; Tom Sesay; James Squire, MPhil; Joseph Kamanda Sesay; Osman Kaykay, MMed; Binyam Halu, MPH; Hailemariam Legesse, Postgraduate Diploma in Paediatrics and Child health; Francis Smart; Sartie Kenneh; Soter Ameh, PhD.

The findings and conclusions in this report are those of the authors and do not necessarily represent the official position of the US Centers for Disease Control and Prevention.

## Author contributions

**Conceptualization:** Portia Chipo Mutevedzi, Zachary J. Madewell, Victor Akelo, Cynthia G. Whitney, Dianna M. Blau, Inacio Mandomando.

**Data curation:** Portia Chipo Mutevedzi, Karen L. Kotloff, Quique Bassat, Percina Joao Chirinda, Anelsio C A Cossa, Elisio G. Xerinda, Victor Akelo, Paul K. Mitei, Elizabeth Oele, Richard Omore, Dickens Onyango, Joseph Bangura, Ronita Luke, Andrew Moseray, Ikechukwu Udo Ogbuanu, Tom Sesay, Nega Assefa, Temesgen Teferi Libe, Lola Madrid, Melisachew M. Yeshi, J. Anthony G. Scott, Nelesh P. Govender, Sanjay G. Lala, Shabir A. Madhi, Sana Mahtab, Adama Mamby Keita, Doh Sanogo, Samba O. Sow, Milagritos D. Tapia, Shams El Arifeen, Emily S. Gurley, Beth A. Tippett Barr, Dianna M. Blau, Inacio Mandomando.

**Formal analysis:** Zachary J. Madewell.

**Funding acquisition:** Cynthia G. Whitney, Dianna M. Blau.

**Investigation:** Portia Chipo Mutevedzi, Zachary J. Madewell, Karen L. Kotloff, Quique Bassat, Percina Joao Chirinda, Anelsio C A Cossa, Elisio G. Xerinda, Victor Akelo, Paul K. Mitei, Elizabeth Oele, Richard Omore, Dickens Onyango, Joseph Bangura, Ronita Luke, Andrew Moseray, Ikechukwu Udo Ogbuanu, Tom Sesay, Nega Assefa, Temesgen Teferi

Libe, Lola Madrid, Melisachew M. Yeshi, J. Anthony G. Scott, Nelesh P. Govender, Sanjay G. Lala, Shabir A. Madhi, Sana Mahtab, Adama Mamby Keita, Doh Sanogo, Samba O. Sow, Milagritos D. Tapia, Shams El Arifeen, Emily S. Gurley, Beth A. Tippett Barr, Cynthia G. Whitney, Dianna M. Blau, Inacio Mandomando.

**Methodology:** Portia Chipo Mutevedzi, Zachary J. Madewell, Karen L. Kotloff, Quique Bassat, Percina Joao Chirinda, Anelsio C A Cossa, Elisio G. Xerinda, Victor Akelo, Paul K. Mitei, Elizabeth Oele, Richard Omore, Dickens Onyango, Joseph Bangura, Ronita Luke, Andrew Moseray, Ikechukwu Udo Ogbuanu, Tom Sesay, Nega Assefa, Temesgen Teferi Libe, Lola Madrid, Melisachew M. Yeshi, J. Anthony G. Scott, Nelesh P. Govender, Sanjay G. Lala, Shabir A. Madhi, Sana Mahtab, Adama Mamby Keita, Doh Sanogo, Samba O. Sow, Milagritos D. Tapia, Shams El Arifeen, Emily S. Gurley, Beth A. Tippett Barr, Cynthia G. Whitney, Dianna M. Blau, Inacio Mandomando.

**Project administration:** Portia Chipo Mutevedzi, Zachary J. Madewell, Karen L. Kotloff, Quique Bassat, Victor Akelo, Ikechukwu Udo Ogbuanu, Nega Assefa, Lola Madrid, J. Anthony G. Scott, Shabir A. Madhi, Shams El Arifeen, Emily S. Gurley, Cynthia G. Whitney, Dianna M. Blau, Inacio Mandomando.

**Supervision:** Karen L. Kotloff, Quique Bassat, Victor Akelo, Ikechukwu Udo Ogbuanu, Nega Assefa, Lola Madrid, J. Anthony G. Scott, Shabir A. Madhi, Shams El Arifeen, Emily S. Gurley, Cynthia G. Whitney, Dianna M. Blau, Inacio Mandomando.

**Validation:** Zachary J. Madewell.

**Visualization:** Portia Chipo Mutevedzi, Zachary J. Madewell.

**Writing – original draft:** Portia Chipo Mutevedzi, Zachary J. Madewell, Victor Akelo.

**Writing – review & editing:** Portia Chipo Mutevedzi, Zachary J. Madewell, Karen L. Kotloff, Quique Bassat, Percina Joao Chirinda, Anelsio C A Cossa, Elisio G. Xerinda, Victor Akelo, Paul K. Mitei, Elizabeth Oele, Richard Omore, Dickens Onyango, Joseph Bangura, Ronita Luke, Andrew Moseray, Ikechukwu Udo Ogbuanu, Tom Sesay, Nega Assefa, Temesgen Teferi Libe, Lola Madrid, Melisachew M. Yeshi, J. Anthony G. Scott, Nelesh P. Govender, Sanjay G. Lala, Shabir A. Madhi, Sana Mahtab, Adama Mamby Keita, Doh Sanogo, Samba O. Sow, Milagritos D. Tapia, Shams El Arifeen, Emily S. Gurley, Beth A. Tippett Barr, Cynthia G. Whitney, Dianna M. Blau, Inacio Mandomando.

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
