## [Decision Letter · Decision Letter 0]

PGPH-D-24-02973

Use of Minimally Invasive Tissue Sampling to determine the contribution of diarrheal diseases to under-five mortality and associated co-morbidities and co-infections in children with fatal diarrheal diseases in Africa and Bangladesh

Dear Dr. Madewell,

Thank you for submitting your manuscript to PLOS Global Public Health. After careful consideration, we feel that it has merit but does not fully meet PLOS Global Public Health’s publication criteria as it currently stands. Therefore, we invite you to submit a revised version of the manuscript that addresses the points raised during the review process.

We look forward to receiving your revised manuscript.

Kind regards,

Espoir Bwenge Malembaka, MD

Academic Editor

Journal Requirements:

1. Please report in the Methods section the dates when data were accessed for research purposes.

Additional Editor Comments (if provided):

1. General comment on added value:

We thank the authors for their important work. As a clinician working in a referral hospital in a low- and middle-income setting, I read this paper with great interest. However, I was left with a sense of "déjà vu". It remains unclear what the real added value of this substantial effort is to the existing body of knowledge on the causes of death and diarrhea in children under five in LMICs. I recommend that the introduction and discussion be strengthened by presenting a clearer and more compelling rationale for the study. Specifically, the authors should articulate what gap this study fills and how it builds upon or differs from previous work in this area.

2. Actionability and policy relevance of findings:

I was encouraged by the first sentence of the final paragraph: “Line 509: Our data provides valuable insights that are translatable into interventions to reduce diarrheal deaths and ultimately reduce under-five mortality.” However, I expected the authors to elaborate further on the policy and programmatic implications of their findings. How do they envision these results informing action, both locally and globally? What changes—at the level of health policy, clinical practice, or public health interventions—do the findings support? In global health, research must lead to clear, actionable recommendations to be truly impactful.

3. HIV and comorbidities:

HIV remains a major contributor to both child malnutrition and sepsis in LMICs. Did the authors examine these comorbidities—particularly HIV—either separately or in combination, and assess their interactions with diarrheal mortality? Exploring such comorbid conditions could offer additional insights, especially given the known syndemic interactions in undernourished pediatric populations.

Reviewers' comments:

Reviewer's Responses to Questions

**Comments to the Author**

1. Does this manuscript meet PLOS Global Public Health’s publication criteria?

Reviewer #1: Yes

2. Has the statistical analysis been performed appropriately and rigorously?

Reviewer #1: Yes

3. Have the authors made all data underlying the findings in their manuscript fully available (please refer to the Data Availability Statement at the start of the manuscript PDF file)?

Reviewer #1: Yes

4. Is the manuscript presented in an intelligible fashion and written in standard English?

Reviewer #1: Yes

Reviewer #1: This is a well written and clear paper. I conclusion are supported by the data presented and by the conclusion of multiple antecedent studies. This manuscript offers an important lens on diarrheal mortality and will be a good addition to the literature. I have a few relatively minor comments:

The role of CT value on the qPCR was unclear to me. Were all levels of detection considered a possible cause of the diarrhea, or did the pannel consider CT value in their attribution?

I do not believe that The EAEC results is in line with previous observations from GEMS and VIDA and MALED. There is a lot of EAEC around, and it is often associated with malnutrition, however it is very surprising to see it come out as the leading cause of diarrhea mortality. Is this result just confounded by nutritional status? Could this reflect a bias in the pannels mortality attribution? A richer discussion of this results is important, particularly as you have GEMS/VIDA investigators on this paper.

Abstract: Why is Rota virus disaggregated in the abstract but no other pathogens are? Surely this just serves to demote Rota in the order of importance. It is helpful to disaggregate it in the manuscript but perhaps not in the abstract.

Sever wasting has been defined by WLZ – are you able to include Kwashikor and MUAC in that definition for the full SAM definition?

Line 319-325 is pretty hard to understand, but I think is making the point that children with sepsis do not die of diarrhea. This is a very convoluted way of making a point that broadly understood already.

Line 467 says “for”, I think should be “or”

**Do you want your identity to be public for this peer review?** For information about this choice, including consent withdrawal, please see our Privacy Policy

Reviewer #1: No

---

## [Editor Report · Decision Letter 1]

Use of Minimally Invasive Tissue Sampling to determine the contribution of diarrheal diseases to under-five mortality and associated co-morbidities and co-infections in children with fatal diarrheal diseases in Africa and Bangladesh

PGPH-D-24-02973R1

Dear Dr. Madewell,

We are pleased to inform you that your manuscript 'Use of Minimally Invasive Tissue Sampling to determine the contribution of diarrheal diseases to under-five mortality and associated co-morbidities and co-infections in children with fatal diarrheal diseases in Africa and Bangladesh' has been provisionally accepted for publication in PLOS Global Public Health.

Best regards,

Espoir Bwenge Malembaka, MD

Academic Editor